# Surgically Induced Demyelination in Rat Sciatic Nerve

**DOI:** 10.3390/brainsci13050754

**Published:** 2023-05-03

**Authors:** Rahul Shankar Rao Rayilla, MUR Naidu, Phanithi Prakash Babu

**Affiliations:** 1Department of Biotechnology and Bioinformatics, School of Life Sciences, University of Hyderabad, Hyderabad 500046, India; 2Department of Pharmacology and Therapeutics, Nizam Institute of Medical Sciences, Hyderabad 500082, India

**Keywords:** sciatic nerve injury, demyelination, teriflunomide, remyelination

## Abstract

Demyelination is a common sign of peripheral nerve injuries (PNIs) caused by damage to the myelin sheath surrounding axons in the sciatic nerve. There are not many methods to induce demyelination in the peripheral nervous system (PNS) using animal models. This study describes a surgical approach using a single partial sciatic nerve suture to induce demyelination in young male Sprague Dawley (SD) rats. After the post-sciatic nerve injury (p-SNI) to the sciatic nerve, histology and immunostaining show demyelination or myelin loss in early to severe phases with no self-recovery. The rotarod test confirms the loss of motor function in nerve-damaged rats. Transmission electron microscopic (TEM) imaging of nerve-damaged rats reveals axonal atrophy and inter-axonal gaps. Further, administration of Teriflunomide (TF) to p-SNI rats resulted in the restoration of motor function, repair of axonal atrophies with inter-axonal spaces, and myelin secretion or remyelination. Taken together, our findings demonstrate a surgical procedure that can induce demyelination in the rat sciatic nerve, which is then remyelinated after TF treatment.

## 1. Introduction

Demyelination is the abnormal process in which the myelin sheath, which protects the axons for proper neural transmission or neuronal communication, is lost or damaged. Myelin loss compromises both axon survival and the flow of electrical impulses in local circuits. Clinical disability arises as a result over time [1]. Diseases like Guillain–Barre syndrome, persistent inflammatory demyelinating polyneuropathy, and injuries can all result in demyelination. 

To date, many methods have been used to cause demyelination, which includes chemically induced demyelination such as cuprizone [2], focal lysolecithin injections [3], tellurium–endoneurial injection to the tibial nerve [4], ethidium bromide–rat sciatic nerve [5], lysophosphatidylcholine–sciatic nerve, virus-triggered spinal cord demyelination, and intracranially injected Theiler’s encephalomyelitis virus [6]. Neuro-inflammatory animal models such as experimental autoimmune encephalomyelitis [7], intravitreal injection of the cytokines (TNFα, TNF/3, IL-1, IL-6, Interferon) [8], and chronic nerve compression model [9] are well known. The cuff model is reported to study allodynia and neuropathic pain [10]. 

Axonotmesis, neurotmesis, and neuropraxia are the three subtypes of PNIs. Level I neuropraxia, which is most frequently observed in wrist drops and muscle twists, is a form of nerve damage or injury that does not include any nerve degeneration. In some rare instances, medication is necessary, but self-healing is generally evident. Neurotmesis is a third-level advanced injury that necessitates surgery yet shows inadequate healing because both the nerve and the nerve sheath are damaged. Axonotmesis, a form of Wallerian degeneration, results in level II nerve injury. It is frequently observed in crash injuries and accident cases wherein the axon and myelin sheath are damaged, but endoneurium, perineurium, and epineurium are remain intact.

TF is an oral immunomodulatory approved drug for the treatment of relapsing-remitting CNS-based demyelinating illnesses such as multiple sclerosis [11,12]. TF suppresses dihydroorotate dehydrogenase, which inhibits pyrimidine production and promotes oligodendroglial differentiation and myelination. A small dose of TF causes the cell cycle to exit, whereas a larger dose causes a reduction in cell survival [13].

In view of this, the current study was designed to induce a sciatic nerve injury rat model using a single partial nerve ligation or suture procedure and to evaluate/examine the phases of demyelination, as well as axonal atrophies and inter axonal gaps, in experimental rats based on prolonged-time periods after p-SNI. In addition, we investigated motor functioning in experimental rats. Further, TF was repurposed to improve myelin restoration for p-SNI rats. 

## 2. Materials and Methods

### 2.1. Ethics Statement

All the animal experiments were carried out after approval from the institutional animal ethical committee (UH/IAEC/PPB/2022/12), University of Hyderabad, India. As per the experimental requirement, animals were acclimatized in the animal house facility, University of Hyderabad, 10 days before the experiments (Reg number: 151/1999/CPCSEA). All the rats used in the study were under complete veterinarian observation before and throughout the experimental procedures.

### 2.2. Animals 

Sprague Dawley (SD) male rats aged (8-week-old) and weighing 220–250 g were purchased from the National Institute of Nutrition (NIN) in Hyderabad, Telangana State, India. The animals were housed in ventilated cages with an ambient temperature of 24 °C, consistent and standard air humidity controlled (21 ± 3 °C, 50 ± 10%), natural day/night cycles, and quality food and water ad libitum. All the animals used in this study are healthy and are not genetically modified. These animals, which are employed in this work, have no prior research background. Every attempt was made to reduce the number of animals used and the suffering they endured. Table 1, Table 2 and Table 3 contains a list of the animals used in various experiments. This study used a sample size of *n* = 6/condition since the entire experiment was conducted on SD-rats, and *n* = 6 is the typical sample size for rat models. Based on the experimental design, rats were randomly divided into *n* = 6/condition. To avoid surgical failures, all surgical procedures on rats were performed by a single expert. All of the animal groups included in this study were first trained on rotarod according to the procedure, and then the sciatic nerve was surgically injured. Afterward, motor function was checked on the rotarod test, and timings were recorded. The same animals were slaughtered, and sciatic nerves were utilized for pathological studies to prevent excessive animal use. 

### 2.3. Animal Groups

The sham group (SH) that had surgery without damaging the sciatic nerve served as a control. The comparison of a sham group to experimental animal groups that underwent partial sciatic nerve ligation, euthanasia after post-sciatic nerve injury (p-SNI) at prolonged-time periods, i.e., experimental groups: rats euthanized on 4th day after p-SNI (4 d), euthanized on 7th day after p-SNI (7 d), euthanized on 10th day after p-SNI (10), and the self-recovery group (SR), which had a partial ligation removed on 10th day after p-SNI and rats were maintained for two weeks or 14 days without treatment to evaluate self-healing. All of the aforementioned animal groups underwent the rotarod test to evaluate the motor functionality before the rats were euthanized. The vehicle group (VEH) represents the oral administration of carboxymethylcellulose prepared at a concentration of 0.06% (*w/v*) in water, to which Tween-80 was added to obtain a final dosage of 0.5% (*v/v*) to sham or control rats to rule out any deleterious effects. The vehicle group was compared to the previously mentioned 10 d p-SNI group, as well as Teriflunomide (TF)-treated animal groups that had their partial ligation removed after 10 days of p-SNI and received their first dose of TF by oral administration immediately after recovery from anesthesia, which was continued for two weeks or 14 days on alternate days. After euthanasia, all the sciatic nerves were collected for further histology, immunostaining, and transmission electron microscopy imaging. For quantification and statistical analysis, each animal group used in this study contained n = 6/condition, i.e., n(SH) = 6, n(4 d) = 6, n(7 d) = 6, n(10 d) = 6, n(SR) = 6, n(VEH) = 6, and n(TF) = 6.

### 2.4. Surgical Procedure

All of the experimental rats were weighed and anesthetized intraperitoneally using ketamine/xylazine as a sedative agent, approximately 60 mg/10 mg per kg body weight of rat [14]. Antibiotics: Gentamycin 8 mg/250 g of SD rat model (only a single dose after performing the surgery). After sedation, the hair at the femoral site was shaved with a surgical blade in the operated region. The rat was placed on the hot plate to keep warm, and the temperature was maintained at 37 °C. For a sterile environment, while performing the surgery, the rat’s femoral region and the surrounding area were cleaned down with 70% ethanol. In order to locate the sciatic nerve, skin and gluteal muscle incisions were made. The sciatic nerve was exposed using forceps, and nerve injury was induced by inserting the 6-0 monofilament thread from the middle portion of the sciatic nerve for partial ligation to promote demyelination. Following injury, 3-0 surgical sutures were used to close the incisions made in the gluteal muscle and skin. For topical anesthesia, Lignocaine hydrochloride gel (2%) was applied on the surface of the operated region and shifted to the recovery chambers for acclimatization as per experimental schedules. The study included animals that underwent appropriate partial ligation of the rat sciatic nerve. If the suture was not correctly passed from the sciatic nerve for partial ligation, the animals were excluded. A few exclusions were made because of improper tissue processing, which might cause issues during pathological evaluations. 

### 2.5. TF Administration

After 10 d p-SNI, rats were sedated, and the partial ligation or suture was removed, and the rats were given their TF doses after recovery from anesthesia (TF provided by NATCO Pharma Limited, Hyderabad, T.S., India). TF was administered orally at a dosage of 10 mg/kg body weight through rat oral gavage for two weeks on alternate days [15]. TF was dissolved in the vehicle: carboxymethylcellulose made up to 0.06% (*w/v*) in water, to which Tween 80 was added to reach a final concentration of 0.5% (*v/v*). Rats were euthanized shortly after the therapy was completed, and sciatic nerves were severed and preserved in 10% formalin for pathological examinations. 

### 2.6. Euthanasia Followed by Nerve Harvesting Procedure

All animals in this study were euthanized in accordance with the guidelines of the Institutional Animal Ethical Committee (approval number: UH/IAEC/PPB/2022/12). Euthanasia: Overdose of sodium pentobarbital (150 mg/kg) was delivered intraperitoneally to sacrifice the animals. All the experimental rats were euthanized and placed on the operation table. Skin and gluteal muscle incisions were performed to remove the operated sciatic nerve from the rats. The harvested nerves were then fixed in 10% formalin for pathological examinations.

### 2.7. Paraffin-Embedded Tissue Sections

Freshly obtained sciatic nerve samples were fixed for 48 h at room temperature in 10% formalin. After fixation, nerve samples were washed for 1 h under running tap water, followed by dehydration phases of 30 min each with 70%, 80%, and 95% alcohol changes, and three changes of 100% alcohol each 1 h. Tissue samples were cleaned in one change of xylene for 5 min, followed by another step of xylene and melted paraffin 1:1 ratio for 5 min, and the tissue was embedded in a paraffin block. Tissue blocks were mounted to the microtome (LEICA RM2145), and 10 µm cross-sections were cut and floated in a 40 °C maintained water bath containing clear distilled water. Sciatic nerve cross slices were then carefully transferred to glass slides for further pathological investigation.

### 2.8. Haematoxylin and Eosin Staining (H & E) 

Tissue cross-sections were deparaffinized twice, for 2 min each, in xylene, and then rehydrated twice, for 3 min each, in 100%, 95%, 80%, and 70% alcohol. Slides were applied for a 10 min water wash, had their nuclei stained with haematoxylin for 3 to 5 min, and then stained with eosin for the extracellular matrix and their cytoplasm for 2 min before being rinsed with tap water for 6 min. Sections underwent rehydration by being exposed to 30% and 50% alcohol for 6 min each, 70% alcohol for 10 min, 95% alcohol, and twice 100% alcohol for 3 min. After cleaning the tissue sections with two changes of xylene for 3 min each, the tissue sections were mounted on DPX slides for imaging under light microscopy.

### 2.9. Eriochrome (Solo Chrome) Cyanine R Myelin Staining (EC)

In the EC myelin stain, the extracellular matrix and cytoplasm are grey and cream, whereas myelin is represented by a blue color. Sections were deparaffinized in two changes of xylene for 3 min each, then rehydrated with two changes of 100% alcohol for 3 min each, then three changes of 95%, 80%, and 70% alcohol for 3 min each. After being rehydrated, slides were transferred for a 10 min water wash. Slides were then immersed in an eriochrome cyanine solution (eriochrome cyanine RS 0.2 g, H2SO4 0.5 mL, distilled water 96 mL, and 10% iron alum 4 mL for 100 mL solution) for 30 min at room temperature before being washed under running water. After being differentiated in 5% iron alum for 5–15 min, the tissue sample was washed with tap water. Borax ferricyanide separation was performed for 5–10 min (borax 1.0 g, potassium ferricyanide 1.25 g, and distilled water 100 mL protected from sunlight) before being washed with running tap water. The slides were then dehydrated with two changes of graded ethanol solutions (70%, 80%, 95%, and 100% alcohol) and cleaned with three changes of xylene before being DPX mounted for microscopy imaging.

### 2.10. Luxol Fast blue Myelin Staining (LFB)

Sciatic nerve cross-sections were deparaffinized in two changes of xylene for 2 min each, then hydrated in 80% and 95% graded ethanol solutions for 3 min each. Slides were maintained in 0.1% luxol fast blue solution (catalog no. L0294, SIGMA ALDRICH^®^) (luxol fast blue solution—0.1 mL, ethyl alcohol 95%—100 mL, glacial acetic acid—0.5 mL) overnight at 56 °C in the oven. The excess stain was removed with 95% ethyl alcohol, followed by a distilled water wash. To differentiate the slides, a 0.05% lithium carbonate solution (catalog no. 431559, 99.99% SIGA ALDRICH^®^) (lithium carbonate 0.05 g in 100 mL of distilled water) was used for 30—45 sec, followed by a 30 sec washing with 70% ethyl alcohol. This is followed by washing in distilled water and counterstained with 0.1% cresyl violet solution (cresyl fast violet—0.1 g in 100 mL of distilled water and 10 drops of glacial acetic acid added shortly before use and filtered) for 30–45 s. Rinse in distilled water, then separate in 95% ethyl alcohol for 5 min, followed by two changes in 100% alcohol for 5 min. The tissue slices were then cleaned in two changes of xylene for 3 min each, followed by DPX mounting for microscope imaging.

### 2.11. Immunostaining

Slides were deparaffinized in two changes of xylene for 2 min each, then promptly transferred to a 1:1 xylene: 100% ethanol solution for 3 min. The slides were rehydrated in graded ethanol solutions of 100%, 95%, 70%, 50%, and 30% ethanol changes for 3 min each before being washed in tap water for 5 min. For antigen retrieval, tissue sections were immersed in an antigen retrieval buffer (trypsin 0.05% in 100 mL PBS) for 15 min at 37 °C and then permeabilized (triton x—100 0.2% in PBS) for 7–10 min, followed by one wash in PBS for 5 min. The slides were placed in a blocking solution (BSA 1%, NGS 5% in PBS) for 1 h at room temperature. Approximately 100 µL of diluted primary antibodies myelin basic protein MBP (AB clonal Catalog No: A1664, 86 Dr. Woburn, MA 01801, United States), S100 Beta EP 32 (PathnSitu Catalog No: CR070–0.1 mL Concentrated, USA-Registered Office 538, Selby Ln, Livermore, CA-94551 USA), and CD68 KP 1 (PathnSitu Catalog No: PM113–0.1 mL concentrated, USA-Registered Office 538, Selby Ln, Livermore, CA-94551 USA (Dilution: 1:200 PBS with 0.02% sodium azide, 50% glycerol, pH 7.3) were applied to the tissue cross-sections and incubated for 30 min at room temperature in a humidified environment. Sections were transferred for two wash steps with PBS for 5 min. Approximately 100 µL of diluted biotinylated secondary antibody was added to the sections on the slides (using the antibody dilution buffer) and incubated in a humidified chamber at room temperature for 30 min (covered from sunlight), followed by 5 min washes with PBS for two times. To show the color of the antibody staining, a DAB substrate solution (freshly produced before use: 0.05% DAB, 0.015% H2O2 in PBS) was applied to the sections on the slides. The necessary color intensity was achieved after less than 5 min of color development, which was followed by three PBS changes that lasted 2 min each. Slides were submerged in haematoxylin for 2–3 min, depending on the thickness of the tissue sections, for counterstaining. This was followed by a 15–20 min water wash. In graded alcohol solutions of two changes of 95% and 100% alcohol, tissue slides were dehydrated for 5 min each. Coverslips were used to protect the mounted tissue sections after applying DPX mounting solution.

### 2.12. Transmission Electron Microscopic Imaging (TEM)

The nerve samples were first fixed in 2.5–3% glutaraldehyde in 0.1 M phosphate buffer saline PBS (pH 7.2) for 24 h at 4 °C, and then they were further fixed in 2% aqueous osmium tetroxide in the same buffer for 2 h. Samples were dehydrated in a series of graded alcohols, penetrated with Araldite 6005 resin, or implanted in spur resin [16]. A glass knife and an ultra-microtome (Leica Ultra cut UCT-GA-D/E-I/100) were used to slice material into extremely thin (50–70 nm) sections, which were then mounted on copper grids and stained with saturated aqueous acetate and Reynolds lead citrate. Samples were examined under TEM in higher magnification and resolution (JEOL-JEM-F200 transmission electron microscopic imaging).

### 2.13. Rotarod Test

All the animal groups (*n* = 6/condition) were trained for one week on the rotarod apparatus before the surgical injury to the sciatic nerve (animal groups: SH, VEH, 4 d, 7 d, 10 d, SR, and TF). To ensure sterility, the device was wiped down with 70% alcohol after each trial. Rats were trained four times a day for 3 min each against a spinning rod at a speed of 5–25 rpm. The mean delay of the rat fall off the revolving rod was measured in sec (s) and recorded for each rat in the range of 120 sec above. Following the p-SNI, all the experimental rats were given the opportunity to undergo a rotarod test for motor-functional examination at prolonged-time periods after p-SNI.

### 2.14. Statistical Analysis

Statistical estimations were quantified and noted as the mean percentage of expression by *ImageJ* software. *Sigma Plot* 11.0 software single group one sample t-test is used to estimate the mean ± SEM. GraphPad Prism 8.0.2, column analysis, and a one-way ANOVA test is used for the multiple comparisons (Tukey test) of the mean and significant *p* value.

## 3. Results

### 3.1. The Surgical Approach for Inducing Demyelination in the Rat Sciatic Nerve

Ketamine/Xylazine 60 mg/10 mg per kg body weight was used to anesthetize rats (intraperitoneally). Fur was trimmed, skin and gluteal muscle incisions were made, and the sciatic nerve was exposed. A 6-0 monofilament thread was implanted from the middle of the sciatic nerve and partially ligated. Skin and gluteal muscle incisions were closed using a 3–0 monofilament surgical thread (Figure 1).

### 3.2. Pathological Evaluation Demonstrates the Changes in Cell Morphology and Demyelination Phases at Prolonged-Time Periods after p-SNI in Rats

Haematoxylin and eosin (H&E) staining reveal cell morphological alterations in distinct day sites of the injured sciatic nerve. The cell morphological abnormalities in the sciatic nerve cross-sections began promptly at 4 d p-SNI, which progressively enhanced from 7 d to 10 d after p-SNI without displaying any SR even after two weeks after removing partial ligation from 10 d p-SNI rats shown in Figure 2A–E. Myelin stains, EC, and LFB verified demyelination commenced at 4 d and subsequently enhanced at 7 d and 10 d without demonstrating any SR even after two weeks after removing partial ligation from 10 d p-SNI rats (Figure 2). Immunostaining with MBP and S100 myelin markers of p-SNI cross-sections confirm early demyelination or myelin loss began at 4 d, which progressively enhanced from 7 d to 10 d without any SR, as represented and graphically in Figure 3.

Monocyte-derived macrophage activation is critical and crucial in promoting successful regeneration or remyelination by releasing growth factors and crucial in clearing inhibitory myelin debris. We used the CD68 marker to label the macrophages and found spontaneous activation and accumulation of macrophages with a higher expression at 4 d p-SNI cross-sections, which steadily increased from 7 d to 10 d after p-SNI in contrast to the SH group. Interestingly, data revealed a reduced expression of CD68 in SR, as illustrated in Figure 3.

### 3.3. TF Therapy Helps p-SNI Rats by Suppressing Macrophage Activation and Boosting Myelin Secretion or Remyelination

Early observations revealed active macrophages and demyelination at the site of the surgically injured rat sciatic nerve.

Following 10 d p-SNI, the rats were anesthetized, the partial ligation or suture was removed, and the animals were given their TF dosages once they recovered from anesthesia for two weeks alternate days (TF provided by NATCO Pharma Limited, Hyderabad, T.S., India). H&E staining of TF-treated rat sciatic nerve cross-sections shows the repair of cell morphological abnormalities, as illustrated in Figure 4, as compared to the 10 d p-SNI and VEH groups. In contrast to p-SNI and VEH animals, EC and LFB myelin stains demonstrated myelin restoration in TF-treated rat sciatic nerve cross-sections (Figure 4). Immunostaining with CD68 demonstrated reduced macrophages in TF-treated rats’ sciatic nerve cross-sections compared to the experimental animal groups and VEH groups (Figure 5). Immunohistochemistry for MBP and S100 indicated myelin repair or remyelination in TF-treated rat sciatic nerve cross-sections against experimental animal groups and VEH groups, graphically depicted in Figure 5.

### 3.4. Inter Axonal Gaps Were Discovered Using TEM Imaging Following p-SNI, Which Was Repaired by TF Treatment

TEM analysis demonstrated distinct axonal morphology in VEH and 4 d after p-SNI. Atrophic axons with inter axonal gaps were seen in 10 d p-SNI cross-sections, coupled with myelin loss (Figure 6).

In contrast, the healing of inter axonal gaps and recovery from the axonal damage following two weeks of TF therapy on alternate days in the p-SNI rats are depicted in Figure 6.

### 3.5. Rotarod Test Identified the Motor-Functional Deficit in p-SNI Rats That Was Remedied by TF Therapy

To assess the injured rats’ motor functional loss, a rotarod test was performed, and the findings revealed that the experimental animal groups lost motor function based on rod stay (s) from 5 to 25 rpm. The rotarod test detected progressively enhanced motor functional loss based on the prolonged-time period after p-SNI in rats. The experimental rats *n* (4 d, 7 d, and 10 d) = 6/condition were compared to the *n*(SH) = 6. Statistical data analysis identified variations of motor functionality depending on rod stay in seconds (s) from 5 to 25 rpm.

The TF therapy aided recovery and noted reversed motor dysfunction, as measured on the rotarod device using a stay-on rod (s) from 5 to 25 rpm, which is compared to the experimental rats *n*(4 d, 7 d, and 10 d) = 6/condition and *n*(VEH) = 6 visually illustrated in Figure 7.

## 4. Discussion

The current study describes a surgical method for inducing demyelination with a simple partial ligation to the rat sciatic nerve using a 6-0 monofilament thread. The study investigates how long macrophages will be active based on the prolonged-time period after p-SNI. The findings show that early myelin loss or demyelination progresses to greater demyelination with no evidence of self-recovery. Axonal atrophies and inter-axonal gaps were found after varying periods of post-surgical sciatic nerve damage. Moreover, the rotarod test revealed that surgically nerve-injured rats had lost motor functioning.

This research reveals that TF has multifunctional properties in sciatic nerve-injured rats via immunomodulation and myelin secretion or remyelination at the injury site. However, TF is a well-known immunomodulatory medicine that is used to aid in the healing and repair of myelin in multiple sclerosis (MS) patients [11,12,13,15,17].

There are now various animal models in use to induce demyelination in the rat sciatic nerve, each with its own set of limitations and downsides. Virus-induced demyelination [5,6], neuro-inflammatory animal models such as experimental autoimmune encephalomyelitis (EAE) [7], and intravitreal injection of the cytokines [8] are well-established to cause demyelination in animal models with certain limitations. The chronic nerve compression paradigm is widely established for inducing demyelination in animals. However, the disadvantage of this model is that the nerve is squeezed to generate demyelination. This makes remyelination research challenging [9]. The cuff model is used to study allodynia and neuropathic pain [10].

Neurotoxins such as ethidium bromide, Lysolecithin, cuprizone, and tellurium are used to promote demyelination. However, this may result in indiscriminate demyelination [2,3,4]. The dose is based on an animal’s susceptibility and tolerance depending on its body metabolism. However, not all animals respond to the same dosage in the same way, and some may develop resistance to the neurotoxins, making comparison studies and data interpretation within a given experiment challenging. Theiler’s murine encephalomyelitis virus (TMEV) is a neuropathogenic virus that affects both susceptible and resistant mouse models, causing persistent demyelination that starts 2–6 weeks after viral injection [5]. In contrast, demyelination begins shortly after the ligation on the 4 d in our model.

Experimental autoimmune encephalomyelitis (EAE) is commonly used to study the neuropathological features in inflammation, demyelination, and axonal loss conditions. The counter-regulatory mechanisms in the EAE model are involved in the resolution of inflammation along with remyelination [7].

Previous animal model limitations prompted us to establish a model that could induce demyelination in rats immediately after injury. This study demonstrates a single partial sciatic nerve ligation to induce demyelination or myelin loss in rats. Pathological investigations show early, moderate, and advanced stages of demyelination in prolonged-time periods following p-SNI in rats.

Our preliminary findings revealed abnormalities in the cell morphology on staining with H&E that began on 4 d after p-SNI and were steadily enhanced from 7 d to 10 d without any self-recovery. This is in agreement with similar studies using H&E as basic stains to examine cell morphological changes [18,19]. In our study, LFB and EC stains identified well-defined zones of demyelination that began on the 4 d p-SNI and subsequently increased from 7 d to 10 d following p-SNI in rats, and there was no evidence of self-recovery (SR). Previous research used similar dyes to analyze the myelin composition of tissues [20,21,22,23,24,25]. Immunostaining further confirms the activation of macrophages with the CD68 marker and myelin loss or demyelination with MBP and S100 markers in varying time periods after p-SNI in rats. Immunostaining with CD68 detected active macrophages after peripheral nerve injury [26]. S100 is preferentially distributed in myelin-forming Schwann cells [27]. S100A8 and S100A9 are the subunits that initiate early inflammation in injured peripheral nerves [28]. In a study, MBP detected early myelination in the rat brain stem [29], and MBP detected myelination in the second trimester human fetal spinal cord [30]. MBP is a well-known marker to detect demyelination in multiple sclerosis, human auditory nerve, and some other neurodegenerative diseases [31,32,33,34]. We found severe alterations in p-SNI rats, including neuroinflammation followed by monocyte-derived macrophage activation, as revealed by the CD68 marker. Immunostaining verified the presence of activated macrophages using CD68 at 4 d p-SNI, which was subsequently elevated from 7 d to 10 d following p-SNI in rats and was reduced in the self-recovery (SR) rats group. Interestingly, histology and immunostaining findings revealed that TF therapy reduced pro-inflammation by suppressing CD68 expression and improved healing and myelin regeneration or remyelination with enhanced expression of MBP and S100 markers in 10 d p-SNI rats with TF versus experimental animal groups. Similarly, CD68 is employed as a labeling stain to detect macrophage build-up, which aids in regeneration and cell debris clearance [35,36,37]. Findings are in agreement with other publication using TF to differentiate myelin-producing cells in the nervous system and to modulate immune response [13].

By using the rotarod test to analyze motor function, it was discovered that, in comparison to earlier studies [38,39], all experimental rats in this research with p-SNI had little motor functional loss as measured by the length of time-stay on the rod (s) at 5–25 rpm. Stay on rod (s) of rats on the rotarod device increased following TF therapy, indicating motor-functional recovery. The rotarod test on all experimental rats presented data as per rod stay (s) and validated that the present model can produce motor-functional loss that depends on the time frame length of surgical harm, which aids researchers in investigating motor functionalities in the PNS.

In comparison to the VEH and 4 d p-SNI rat groups, TEM data indicate atrophic axons, inter axonal space, and myelin loss in 10 d p-SNI rats. According to prior research, electron-microscopic alterations in a rat’s sciatic nerve following differential traction damage were described [40,41,42]. Significant changes in axonal intactness and repair of inter-axonal gaps were observed in the TF-treated p-SNI rats, supporting our neuropathological findings on demyelination followed by recovery and remyelination after TF therapy.

## 5. Conclusions

In conclusion, the demyelination is accurate and reproducible, and the nerve damage is minimal since a single partial ligation is conducted to target just the portion of the sciatic nerve that contains axons. We observed identical demyelination throughout the whole batch and location, and the amount of ligation may be adjusted, making comparison studies easier. This model is appropriate for studying the molecular process behind demyelination and gives a chance to test possible novel therapeutics for demyelinating illnesses. This rat model may be more valuable to pharma, the biotech industry, and academic researchers in screening newly synthesized drugs.

## Figures and Tables

**Figure 1 brainsci-13-00754-f001:**
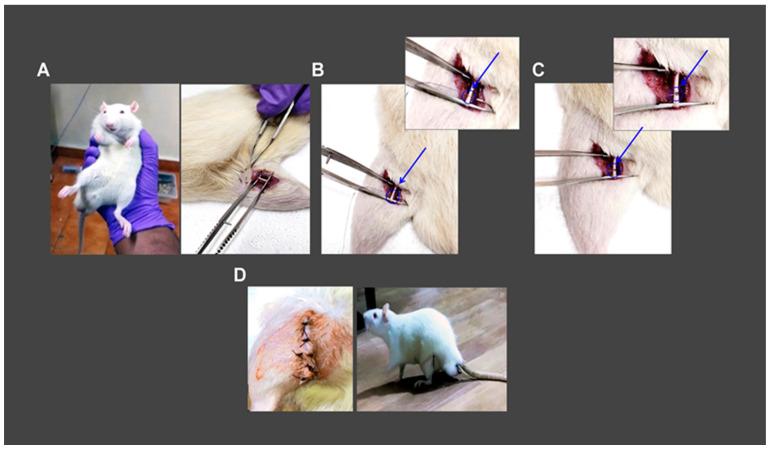
Depicts the surgical method used to injure the rat’s sciatic nerve. (**A)** Rats were sedated, incisions were made in the skin and gluteal muscles, and the sciatic nerve was exposed. (**B**) *Partial* ligation with 6–0 monofilament suture thread placed from the center of the sciatic nerve (blue arrow). (**C**) The sciatic nerve was partly ligated with a 6–0 monofilament thread (blue arrow) to induce myelin loss in the rat’s sciatic nerve. (**D**) Gluteal muscle and skin incisions were closed using a 3–0 polyamide black monofilament thread, and rats recovered after anesthesia.

**Figure 2 brainsci-13-00754-f002:**
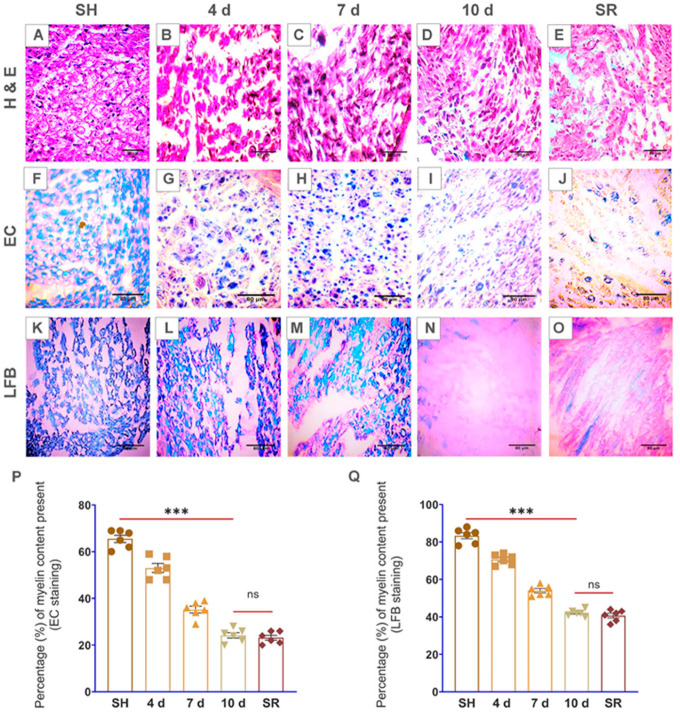
Photomicrographs demonstrate cell morphological changes and demyelination in the p-SNI rat’s sciatic nerve cross-sections. (**A**–**E**) Hematoxylin and eosin staining, which revealed cell-morphological abnormalities, commenced on the *n*(4 d) = 6 and subsequently increased from *n*(7 d) = 6, *n*(10 d) = 6 up to *n* “self-recovery” (SR) = 6 p-SNI rats in contrast to the *n* “sham” (SH) = 6 rats. (**F**–**J**) EC stain (blue color: myelin; cream color: extracellular spaces) and (**K**–**O**) LFB stain (blue color: myelin; pink color: extracellular spaces) demonstrates steadily increasing myelin loss in nerve-damaged groups from the *n*(4 d) = 6, *n*(7 d) = 6, *n*(10 d) = 6 until *n* “self-recovery” (SR) = 6 in contrast to the *n* “sham” (SH) = 6 rats. Statistical information was visually represented in (**P**,**Q**). Olympus BX51 microscope was used to investigate stains and produce photographs. The results were all provided as mean ± SEM. Statistical significance: ns, *p* = 0.988, 0.902, ***, and *p* ≤ 0.001. The scale bar in the images (**A**–**O**) is 80 µm. ImageJ software is used to quantify all of the photos by following measurement settings and analyzing for the mean values. SigmaPlot 11.0 is used to produce statistical results, which are then examined using column analyses, one-way ANOVA, multiple comparisons, and the Tukey test and visually displayed using GraphPad Prism 8.

**Figure 3 brainsci-13-00754-f003:**
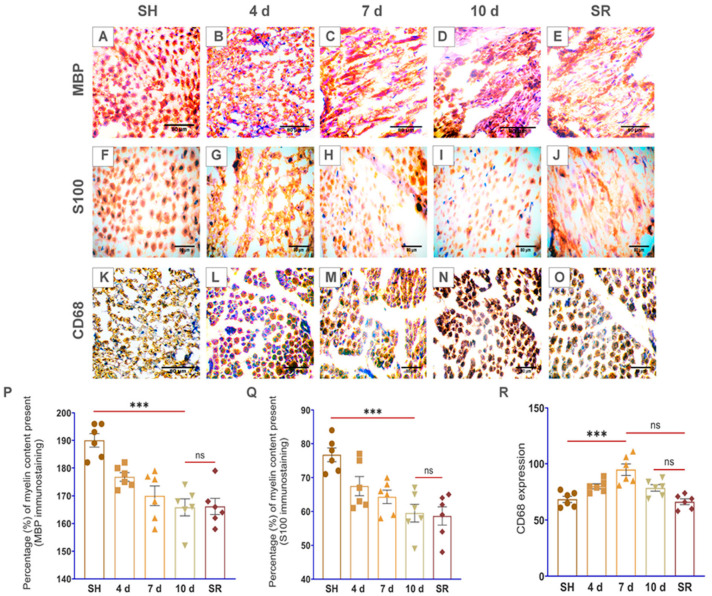
Depicts demyelination phases after p-SNI at prolonged-time periods. Myelin basic protein (**A**–**E**) and S100 (**F**–**J**) were utilized as immunostaining indicators to designate the content of myelin present in nerves, revealing steadily increasing myelin loss in *n*(4 d) = 6, *n*(7 d) = 6, *n*(10 d) = 6, and *n* “self-recovery” (SR) = 6 p-SNI rats when compared to the *n* “sham” (SH) = 6 rats. (**K**–**O**) CD68 marker demonstrated an increase of macrophage accumulation on the *n*(4 d) = 6, which elevated in *n*(7 d) = 6, worsened at *n*(10 d) = 6, and reduced in *n* “self-recovery (SR) = 6 rats when compared to the *n* “sham” (SH) = 6. Statistical data was portrayed graphically in (**P**–**R**). *Olympus BX51* microscope was used to examine the stains and produce photographs. The results were all provided as mean ± SEM. Statistical significance: ns, *p ≥* 0.999, 0.084, and ***, *p* ≤ 0.001. The scale bar in the images (**A**–**O**) is 80 µm. ImageJ software is used to quantify all of the photos by following measurement settings and analyzing for the mean values. SigmaPlot 11.0 is used to produce statistical results, which are then examined using column analyses, one-way ANOVA, multiple comparisons, and the Tukey test and visually displayed using GraphPad Prism 8.

**Figure 4 brainsci-13-00754-f004:**
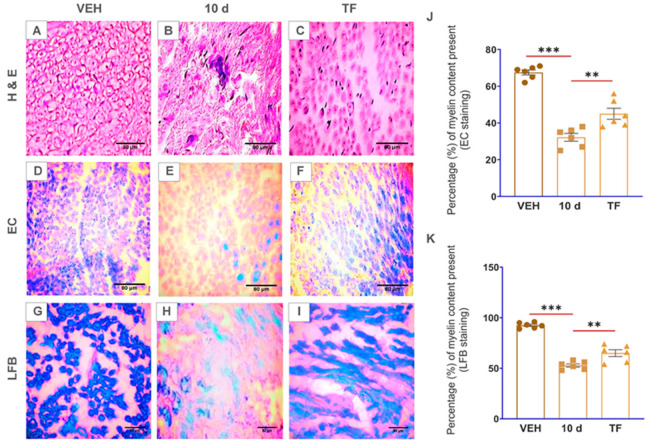
Photomicrographs demonstrate myelin restoration in TF-treated sciatic nerve cross-sections after p-SNI. (**A**–**C**) Haematoxylin and eosin staining revealed the restoration of cell morphological features in *n*(TF) = 6 rats as compared to the *n*(10 d) = 6 and *n* “vehicle” (VEH) = 6. (**D**–**F**) EC stain (blue color: myelin; cream color: extracellular spaces) and (**G**–**I**) LFB stain (blue color: myelin; pink color: extracellular spaces) show remyelination or myelin secretion in *n*(TF) = 6 treated rats when compared to *n*(10 d) = 6 and *n* “vehicle” (VEH) = 6. Statistical data was represented visually in (**J**,**K**). The Olympus BX51 microscope was used to study stains and take images. The results were all provided as mean ± SEM. Statistical significance: **, *p* = 0.003, 0.004, and ***, *p* ≤ 0.001. The scale bar in the images (**A**–**I**) is 80 µm. ImageJ software is used to quantify all of the photos by following measurement settings and analyzing for the mean values. SigmaPlot 11.0 is used to produce statistical results, which are then examined using column analyses, one-way ANOVA, multiple comparisons, and the Tukey test and visually displayed using GraphPad Prism 8.

**Figure 5 brainsci-13-00754-f005:**
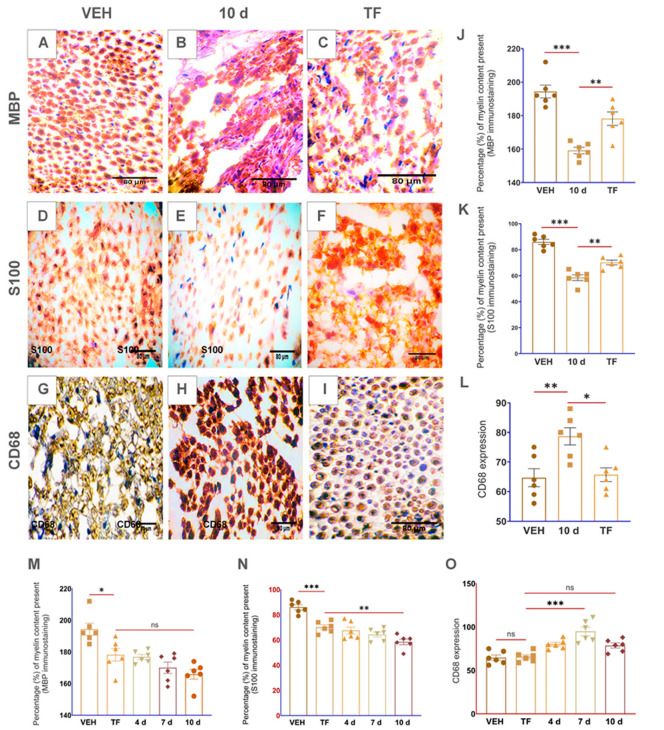
Immunostaining demonstrates myelin repair and macrophage inhibition in TF-treated rat sciatic nerves. When *n*(TF) = 6 rats were compared to *n*(10 d) = 6 and *n* “vehicle” (VEH) = 6 rats, (**A**–**C**) myelin basic protein and (**D**–**F**) S100 myelin indicators revealed myelin restoration or remyelination in *n*(TF) = 6 rats. (**G**–**I**) CD68 demonstrated reduced macrophage accumulation in *n*(TF) = 6 rats when compared to *n*(10 d) = 6. Statistical data was represented visually in (**J**–**O**). The Olympus BX51 microscope was used to study stains and take images. The results were all provided as mean ± SEM. Statistical significance: *, *p* = 0.12, **, *p* = 0.003, and ***, *p ≤* 0.001. The scale bar in the images (**A**–**I**) is 80 µm. ImageJ software is used to quantify all of the photos by following measurement settings and analyzing for the mean values. SigmaPlot 11.0 is used to produce statistical results, which are then examined using column analyses, one-way ANOVA, multiple comparisons, and the Tukey test and visually displayed using GraphPad Prism 8. For TF group quantification comparison, Figure 3 was utilized to get quantitative data for the distinct periods following p-SNI (4 d, 7 d, and 10 d).

**Figure 6 brainsci-13-00754-f006:**
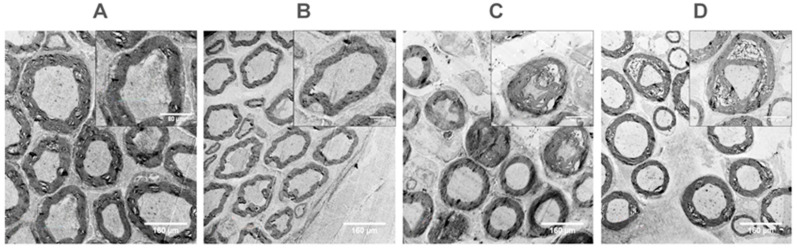
Indicate the restoration of axonal intactness and inter-axonal gaps after TF administration using transmission electron microscopy. (**A**,**B**) Shows clear axonal morphology in the *n* “vehicle” (VEH) = 6 and *n*(4 d) = 6 p-SNI rats. (**C**) Displays the alterations in axonal morphology and interaxonal gaps in the *n*(10 d) = 6 p-SNI rats. (**D**) Shows that the *n*(TF) = 6 rats’ axonal morphology and interaxonal spacing have been restored. JEOL-JEM-F200 transmission electron microscopy (TEM) was used to collect all of the images. The scale bar in the images (**A**–**D**) is 160 µm.

**Figure 7 brainsci-13-00754-f007:**
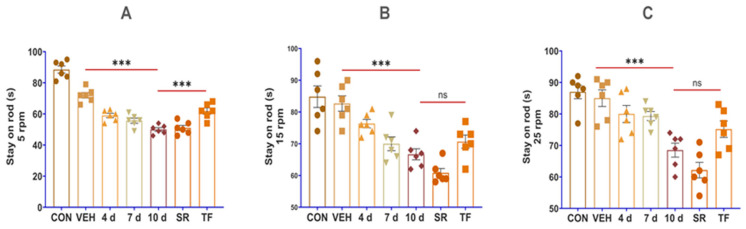
Depict that TF therapy restored motor function following surgical damage to the sciatic nerve. (**A**) 5 rpm, (**B**) 15 rpm, and (**C**) 25 rpm, graphical illustration of rotarod test statistical analyses demonstrates a progressive rise in motor-functional loss from the *n*(4 d) = 6, *n*(7 d) = 6, *n*(10 d) = 6, to *n* “self-recovery” (SR) = 6 p-SNI rats when compared with *n* “control” (CON) = 6 and *n* “vehicle” (VEH) = 6 rats. In the *n*(TF) = 6 rats, motor functioning was restored. The stay-on rod (s) seconds at 5–25 rpm were used for all statistical assessments. The results were all provided as mean ± SEM. Statistical significance: ns, *p =* 0.431, 0.847, and ***, *p * ≤ 0.001. SigmaPlot 11.0 is used to produce statistical results, which are then examined using column analyses, one-way ANOVA, multiple comparisons, and the Tukey test, and graphically displayed using GraphPad Prism 8. Rotarod test speed 5–25 rpm.

**Table 1 brainsci-13-00754-t001:** Experiment 1. Histology, immunostaining, and rotarod test studies to evaluate the phases of demyelination along with motor-functional ability (early to late phases) based on prolonged-time periods after p-SNI to rats.

Animal Group	Species, Age, Weight, and Sex	Number of Animals per Group (*n*)
Sham	Sprague Dawley (SD) male rats, 8-week-old, and weighing 220–250 g	6
4d p-SNI	Sprague Dawley (SD) male rats, 8-week-old, and weighing 220–250 g	6
7 d p-SNI	Sprague Dawley (SD) male rats, 8-week-old, and weighing 220–250 g	6
10 d p-SNI	Sprague Dawley (SD) male rats, 8-week-old, and weighing 220–250 g	6
SR p-SNI	Sprague Dawley (SD) male rats, 8-week-old, and weighing 220–250 g	6

**Table 2 brainsci-13-00754-t002:** Experiment 2. Histology, immunostaining, and rotarod test to evaluate the myelin secretion and motor functionality after TF therapy in p-SNI rats.

Animal Group	Species, Age, Weight, and Sex	Number of Animals per Group (*n*)
Vehicle	Sprague Dawley (SD) male rats, 8-week-old, and weighing 220–250 g	6
10 d p-SNI	Sprague Dawley (SD) male rats, 8-week-old, and weighing 220–250 g	-
TF administered rats group	Sprague Dawley (SD) male rats, 8-week-old, and weighing 220–250 g	6

Note. Experiment 2 utilizes the same 10 d p-SNI group aforementioned in Experiment 1 being compared to Vehicle and TF groups.

**Table 3 brainsci-13-00754-t003:** Experiment 3. TEM imaging was employed to study the axonal alterations in the sciatic nerve following p-SNI and TF treatment.

Animal Group	Species, Age, Weight, and Sex	Number of Animals per Group (*n*)
Vehicle	Sprague Dawley (SD) male rats, 8-week-old, and weighing 220–250 g	6
4 d p-SNI	Sprague Dawley (SD) male rats, 8-week-old, and weighing 220–250 g	6
10 d p-SNI	Sprague Dawley (SD) male rats, 8-week-old, and weighing 220–250 g	6
TF administered rats group	Sprague Dawley (SD) male rats, 8-week-old, and weighing 220–250 g	6

We used a sample size of *n* = 6/condition since the entire experiment was conducted on SD rats, and *n* = 6 is the typical sample size for rat models.

## Data Availability

The data that support the findings of this research are accessible from the corresponding author upon reasonable request.

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
