# Peer review of "Surgically Induced Demyelination in Rat Sciatic Nerve"

_brainsci, 2023, doi:10.3390/brainsci13050754_

Round 1

Reviewer 1 Report

The study used a surgical technique to induce demyelination in the sciatic nerve of young Sprague Dawley rats, resulting in a loss of motor function and axonal atrophy. However, Teriflunomide (TF) treatment restored motor function and led to myelin secretion or remyelination. These findings suggest that this surgical approach can be used to study demyelination and remyelination in the peripheral nervous system and that TF may be a potential therapeutic option for nerve damage. It can be considered published in Brain Sciences with minor revision. Comment as follows:

1.  It seems that surgery skills and acute damage could affect the result. Authors should discuss how to control acute damage caused by animal surgery.

2. n = 6 animals per group were used thought out the studies. Did you calculate the effect size to obtain the sample size?(J Pharmacol Pharmacother. 2013 Oct-Dec; 4(4): 303–306)

3. Please add detail to the “ImageJ software is used to quantify all photos” (Fig.2-5). Please describe how to control the confocal laser intensity and sample variation.  

4. In figure6, please label the zoom-in region in the zoom-out image.

5. Please format the reference.

Author Response

S. No.

Comment

Response

1

It seems that surgery skills and acute damage could affect the result. Authors should discuss how to control acute damage caused by animal surgery.

i. All animal experiments were supervised by an expert with extensive experience in the animal surgery and kept in the care of a Veterinarian.

ii. We agree with the reviewer that the current paradigm generates acute damage. Yet, it has a minor effect on the outcomes since we are just partially ligating the sciatic nerve.

iii. Prolonged time periods in experimental animal groups show a steady rise in myelin loss but not sudden demyelination.

2

n = 6 animals per group were used thought out the studies. Did you calculate the effect size to obtain the sample size? (J Pharmacol Pharmacother. 2013 Oct-Dec; 4(4): 303–306.

Yes, effect size was calculated to obtain the sample size through power analysis by employing G*Power 3.1 software.

(In Experiment 1, the sample size is 25 for 5 groups, with 25/5 = 5 rats in each group. Nonetheless, like in previous studies, we utilized 6 rats per group; in Experiment 2, the sample size is 16 for 2 groups. Each group has 16/2 = 8 rats. Nonetheless, we employed 6 rats per group as per earlier study to maintain animal numbers consistency across all groups; in Experiment 3, the sample size is 24 for 4 groups. The animal number was kept constant (24/4 = 6 rats in each group).

3

Please add detail to the “ImageJ software is used to quantify all photos” (Fig.2-5). Please describe how to control the confocal laser intensity and sample variation.

i. Figure legends 2-5 now provide information on ImageJ software.

ii. We did not use confocal laser-based approaches to data analysis in any experiment. We have used Transmission Electron Microscopy and Light microscopy

4

In figure 6, please label the zoom-in region in the zoom-out image.

We have modified Figure 6 and incorporated it into the revised paper by labelling the zoom-in section of the zoom-out image.

5

Please format reference

References are formatted in revised manuscript.

Reviewer 2 Report

Summary:

The authors described a surgical approach to induce demyelination in the peripheral nervous system of young male Sprague Dawley rats using a single partial sciatic nerve suture. They found that the post-sciatic nerve injury caused demyelination or myelin loss, which was confirmed through histology and immunostaining. The nerve-damaged rats also experienced a loss of motor function, and transmission electron microscopic (TEM) imaging revealed axonal atrophy with inter-axonal gaps. Administration of Teriflunomide (TF) to the injured rats resulted in the restoration of motor function, repair of axonal atrophies, and myelin secretion or remyelination.

General comments:

Overall, the study provides evidence for a surgical method to induce and subsequently remyelinate demyelination in the rat sciatic nerve, using TF treatment. The work is well supported, and the evidence is laid out quite clearly. A few points would need clarification for it to be considered for publication in Brain Sciences.

Specific comments:

The observation of CD68 reduction after TF treatment while remyelination increasing is surprising, I would recommend the authors provide different timepoint of myelination quantification as well as CD68 quantification after TF treatment to fully capture the process.

If this phenomenon persists in the new data, I would recommend the authors to extend the discussion a bit more on this part in the discussion session.

To increase the readability of the paper for the audiences, I recommend the authors list out the full name of the experimental condition in all figure legends (such as XXX (SH), XXX (SR)).

I understand the detail of TF experiment has been provided in the method section, but I would recommend also including some details of the experiment when the authors start to talk about this part in the result session.

Author Response

S. No.

Comment

Response

1

The observation of CD68 reduction after TF treatment while remyelination increasing is surprising, I would recommend the authors provide different timepoint of myelination quantification as well as CD68 quantification after TF treatment to fully capture the process.

i. CD68 expression was enhanced as a result of the prolonged period that addresses pro-inflammation. TF, is an anti-inflammatory compound that suppresses pro-inflammation.

ii. We acknowledge the comment and have incorporated new graphs in the manuscript utilising figure 3 (experimental groups with various timeframes following p-SNI) and figure 5 (Vehicle and TF treated) quantifications.

2

If this phenomenon persists in the new data, I would recommend the authors to extend the discussion a bit more on this part in the discussion session.

We have now modified the discussion part.

3

To increase the readability of the paper for the audiences, I recommend the authors list out the full name of the experimental condition in all figure legends (such as XXX (SH), XXX (SR)).

Reviewers recommendations are appreciated and the changes are incorporated in all figure legends such as n “sham” (SH) = 6, n “self-recovery” (SR) = 6, and n “vehicle” (VEH) = 6.

4

I understand the detail of TF experiment has been provided in the method section, but I would recommend also including some details of the experiment when the authors start to talk about this part in the result session.

A part of TF treatment is added in the results session.